# Exploring the Metabolic Differences between Cisplatin- and UV Light-Induced Apoptotic Bodies in HK-2 Cells by an Untargeted Metabolomics Approach

**DOI:** 10.3390/ijms24087237

**Published:** 2023-04-14

**Authors:** Samuel Bernardo-Bermejo, Elena Sánchez-López, María Castro-Puyana, Ana B. Fernández-Martínez, Francisco Javier Lucio-Cazaña, María Luisa Marina

**Affiliations:** 1Universidad de Alcalá, Departamento de Química Analítica, Química Física e Ingeniería Química, Ctra. Madrid-Barcelona Km.33.600, 28871 Alcalá de Henares (Madrid), Spain; 2Center for Proteomics and Metabolomics, Leiden University Medical Center, Albinusdreef 2, 2333 ZA Leiden, The Netherlands; 3Universidad de Alcalá, Instituto de Investigación Química Andrés M. del Río, Ctra. Madrid-Barcelona Km. 33.600, 28871 Alcalá de Henares (Madrid), Spain; 4Universidad Autónoma de Madrid, Departamento de Biología, Facultad de Ciencias, Campus de Cantoblanco, Calle Darwin, 2, 28049 Madrid, Spain; 5Universidad de Alcalá, Departamento de Biología de Sistemas, Ctra. Madrid-Barcelona Km. 33.600, 28871 Alcalá de Henares (Madrid), Spain

**Keywords:** apoptotic bodies, extracellular vesicles, HK-2 cells, liquid chromatography-mass spectrometry, non-targeted metabolomics

## Abstract

Among the extracellular vesicles, apoptotic bodies (ABs) are only formed during the apoptosis and perform a relevant role in the pathogenesis of different diseases. Recently, it has been demonstrated that ABs from human renal proximal tubular HK-2 cells, either induced by cisplatin or by UV light, can lead to further apoptotic death in naïve HK-2 cells. Thus, the aim of this work was to carry out a non-targeted metabolomic approach to study if the apoptotic stimulus (cisplatin or UV light) affects in a different way the metabolites involved in the propagation of apoptosis. Both ABs and their extracellular fluid were analyzed using a reverse-phase liquid chromatography-mass spectrometry setup. Principal components analysis showed a tight clustering of each experimental group and partial least square discriminant analysis was used to assess the metabolic differences existing between these groups. Considering the variable importance in the projection values, molecular features were selected and some of them could be identified either unequivocally or tentatively. The resulting pathways indicated that there are significant, stimulus-specific differences in metabolites abundancies that may propagate apoptosis to healthy proximal tubular cells; thus, we hypothesize that the share in apoptosis of these metabolites might vary depending on the apoptotic stimulus.

## 1. Introduction

Extracellular vesicles (EVs) are nanometric spheroidal particles which are secreted to the extracellular environment both from apoptotic and healthy cells [1]. EVs perform an important role and open a new frontier in biological processes, such as immune response, signal transduction, or even intracellular communication, in both physiological and pathological conditions, being detected ultimately in several body fluids, such as urine or blood [1]. In this sense, the study of EVs has become a trending topic for biomarker searching [2]. Taking into account their diameter size and density, biogenesis and secretory pathway, EVs can be classified as exomeres (<50 nm), exosomes (30–150 nm), microvesicles (100–1000 nm), and apoptotic bodies (ABs), which are the largest ones (500–2000 nm). Unlike exomes and microvesicles, ABs are only formed during apoptosis, a caspase-dependent mode of programmed cell death [3]. Initially, the only accepted role of ABs was the facilitation of apoptotic cell removal through helping phagocytic cells to engulf smaller ‘bite-size’ fragments of apoptotic cells [4]. More recently, it has been reported the role of ABs in the pathogenesis of various diseases [5]. We have previously found that ABs from human renal proximal tubular epithelial HK-2 cells propagate apoptosis to naïve HK-2 cells independently of the type of the initial apoptotic stimulus (treatment with cisplatin, exposure to UV light or nutrient deprivation) that generated the ABs [6]. In this sense, which is a relevant issue in connection with our previous results, despite the similar degree of apoptosis induced by these different stimuli, there are significant, stimulus-specific differences in metabolites abundancies that may propagate apoptosis to healthy proximal tubular cells so that their role in the apoptosis might vary depending on the apoptotic stimulus.

Among the omics sciences, metabolomics is inherently the closest to phenotype, allowing the characterization of metabolites that translate into the structure, function, and dynamics of an organism [7]. This science is based on powerful advanced analytical techniques where liquid chromatography-mass spectrometry (LC-MS) has proven to offer a high sensitivity and great robustness, what explains why it is the most employed technique in metabolomics [8].

Non-targeted metabolomics approaches have been used for analyzing the metabolome of EVs. This is evidenced by the growing number of works that were published in the last few years [9]. Most of the EVs studies are focused on exosomes from different origin among other EVs of different size. LC-MS and gas chromatography-mass spectrometry were the most used techniques and allowed us to obtain a valuable information to understand the role of EVs in both health and disease. However, it is important to highlight that the applicability of EVs in a clinical setting is still limited because of the lack of standardized protocols [9]. However, as far as we know, non-targeted metabolomics studies exclusively based on ABs from HK-2 cells, have not been reported. In this sense, the purpose of this work was to perform a non-targeted metabolomic study, using a reverse-phase LC-MS (RPLC-MS) setup, on ABs and medium from apoptotic HK-2 cells undergoing apoptosis after treatment with the chemotherapeutic agent cisplatin or exposure to UV light.

## 2. Results and Discussion

### 2.1. Non-Targeted Metabolomics Analysis of ABs Induced by Cisplatin and UV Light Exposure

Since HK-2-cells-generated-Abs, both by cisplatin or upon UV light exposure, can induce apoptosis in HK-2 cells, the purpose of this work was to study their metabolome and therein find the molecular features which help us to unravel the mechanisms of that phenomenon. In this sense, metabolomics is a powerful tool to find differences in metabolite abundances and propose potential deregulated metabolic pathways. To carry out this work, an untargeted metabolomic approach based on RPLC-MS and ESI+ mode, which was developed previously for the analysis of HK-2 cells [10,11], was employed. ABs induced by cisplatin, ABs induced by UV light, and non-treated HK-2 cells were the three sample groups analyzed. In order to obtain a wider point of view, ABs content not only was studied, but also extracellular fluid was analyzed. Each of these two metabolomics sequences consisted of five biological replicates per group (three groups) and each being analyzed three times (analytical replicates) and QCs injections were distributed evenly along the sequence (a total of 15 QCs injections) (see Section 3.5). At the end of the sequence, data obtained were treated following the Section 3.6, and once molecular feature alignment, filtering, and normalization were carried out, the final datasets showed 561 and 479 variables for ABs and extracellular fluids, respectively.

As a consequence of the inherent variability in each sample, the data matrix obtained from the metabolomic sequences is even more complicated. In this sense, multivariate statistical analysis is a helpful option to understand and interpret the data from metabolomics studies. First, principal components analysis (PCA), one of the non-supervised multivariate analyses most frequently used in metabolomics, was performed (Figure 1).

This statistical analysis reduces the dimensionality of the data but preserves the variation present in the original data set as much as possible [12]. In this line, PCA enables us to test the stability of the two sequences and to examine the possible metabolic differences. Previous to the PCA performance, data normalization was a key factor in metabolomic analysis for the data quality and to carry out a correct interpretation of the results. Nevertheless, the normalization methods employed for untargeted metabolomic analysis of EVs are limited and there is not a clear consensus [9]. For example, some of these strategies include protein concentration [13] or particle number [14]. In this way, the method selected in this work was the protein concentration, as detailed in the experimental part. As it can be seen in Figure 1, QC samples appear clustered together in both ABs fluid and extracellular fluid sequences, which showed the high quality and reproducibility of them. The use of QC samples is commonly employed in metabolomics because not only allows to check the system performance, but also to control the equilibration of the analytical system and the signal intensity, among other advantages. On the other hand, a good clustering among the different sample groups can be observed in Figure 1. Particularly in the ABs fluid there was a small, yet noticeable separation between UV- and cisplatin-ABs on the PC1.

### 2.2. Selection of Variables Based on Variable Importance

In order to find the most significant changes between ABs induced by cisplatin and ABs induced by UV light, partial least squares-discriminant analyses (PLS-DA) were performed for both sequences. Unlike PCA, PLS-DA is a supervised multivariate analysis that relies on the class membership of each observation. As Figure 2 shows a good separation between both groups was achieved and the values obtained for the R^2^X, R^2^Y, and Q^2^ parameters demonstrated the quality of these PLS-DA models. To assess the veracity of these models, a cross-validated ANOVA (CV-ANOVA) analysis was carried out. High F-values and low p-values were obtained in both fluids (F-value= 244.0 and *p*-value= 1.2 × 10^−14^ for ABs fluid; F-value= 139.7 and *p*-value= 1.0 × 10^−11^ for extracellular fluid).

To ensure that the separation showed by PLS-DA was not achieved purely by chance, a permutation test using 200 permutations was performed, obtaining positive slopes for both ABs fluid and extracellular fluid (Figure 3), which emphasizes the veracity of the PLS-DA models [15]. Finally, a variable importance in the projection (VIP) value higher than 1.10 was used to select those variables that changed significantly between ABs induced by cisplatin and ABs induced by UV.

### 2.3. Identification of the Selected Molecular Features

Once the statistical analysis was carried out, as common scenario in untargeted metabolomics, the identification of the metabolites was conducted next. According to the procedure explained in Section 3.7, a total of 122 and 112 variables were selected as statistically significant molecular features from ABs fluid and extracellular fluid, respectively. Table 1 shows the retention time, molecular formula, monoisotopic mass, mass error, main MS/MS fragments, VIP, and trend corresponding to those metabolites that were identified as level 1 or level 2 according to the Metabolomics Standard Initiative (MSI) guidelines [16].

In this sense, creatine and spermine (in the ABs fluid), nicotinamide (in the extracellular fluid) and valine, leucine, and 5’-methythioadenosine and pyridoxine (in both fluids) were identified as level 1, i.e., matching retention time, accurate monoisotopic mass, and MS/MS fragments with the standards. N-(1-Deoxy-1-fructosyl)leucine (o isoleucine), N-(1-Deoxy-1-fructosyl)phenylalanine, and N-(1-Deoxy-1-fructosyl)tyrosine were identified in both fluids as level 2, i.e., comparing the accurate mass and fragmentation spectra to available databases such as HMDB or CFM-ID (cfmid.wishartlab.com). Lastly, Appendix A includes the retention time, monoisotopic mass, main fragments, VIP (>2.0), and trend of those metabolites classified as level 4, i.e., the accurate mass was unknown, and any metabolite could be annotated.

### 2.4. Biological Interpretation

There are no references on the apoptotic effects of metabolites contained in ABs because the field of the role of ABs in pathology is in its infancy. To the best of our knowledge, there are only a few references regarding their implication in tumorigenesis [17], autoimmunity [18,19,20], inflammation, and viral infection [21,22]. The aim of this work was to investigate whether in spite of the similar degree of apoptosis induced by different stimuli in HK-2 cells [6], the type of apoptotic stimulus gives rise to significant differences in metabolites (either included into ABs or directly released to the extracellular medium) with potential ability to propagate apoptotic from apoptotic HK-2 cells, so that the apoptotic stimulus might affect in a different way the propagation of apoptosis.

Among the changes potentially relevant are the differences in the content of pyridoxine and kynurenine between the two types of ABs, i.e., cisplatin-treated vs. UV light exposed-cells, both in ABs content (Table 1 and Appendix A) and in the extracellular fluid (Table 1 and Appendix A). Pyridoxine is a major vitamer of vitamin B6 and, because it cannot be synthesized by mammalian cells, it is a normal component of mammalian cell culture media. Pyridoxal kinase converts pyridoxine into pyridoxal 5’-phosphate (the active form of vitamin B6), which is a coenzyme in lipid, amino acid, carbohydrate, and protein metabolism [23]. As the pyridoxine levels in the culture medium from cisplatin-treated cells were lower than those found in medium from UV light-exposed cells (Appendix A) it is likely that an increased uptake may contribute to the higher content in pyridoxine in ABs from cisplatin-treated cells as compared to that found in ABs from UV light-exposed cells (Appendix A). Kynurenine is an early metabolite in the kynurenine pathway for de novo synthesis of NAD+, which metabolizes dietary L-tryptophan to quinolinic acid [24]. We cannot conclude that the differences in kynurenine content is due to a different uptake of tryptophan since this metabolite could not be found in our dataset. Therefore, the more plausible hypothesis is that the divergencies in the content in kynurenine between ABs are most likely secondary to changes in the activity of enzymes of the kynurenine pathway. Since both pyridoxine and kynurenine have been previously involved in noxious cellular effects [25,26,27,28], it is particularly relevant that the content of the two metabolites in both types of ABs was statistically higher than that found in untreated control HK-2 cells. Pyridoxine has been shown to enhance the cytotoxic effect of cisplatin in non-small cell lung carcinoma through increasing its transporter-dependent uptake [25] and to have cytotoxic effects on human fibroblasts in a manner dependent on the photoproducts generated after irradiation of pyridoxine with UV light [26]. In turn, kynurenine, through activation of the aryl hydrocarbon receptor, is known to mediate free radical accumulation and neuronal or cardiomyocyte apoptosis [27]. Additionally, the generation and subsequent accumulation of kynurenine pathway metabolites has been involved in the mechanism of proximal tubular cell death in an experimental model of injury [28]. Due to the differences found in our study between the two different treated ABs, there may be also quantitative differences in the degree in which these metabolites contribute to the induction of apoptosis in naïve HK-2 cells by ABs, depending on the ABs’ origin. However, targeted studies are needed to clarify this point.

Nicotinamide was the only statistically significant variable that could be unambiguously identified in the extracellular medium. Nicotinamide deficiency has been linked to the damage of the renal tubular cells, especially proximal tubular cells. For instance, intraperitoneal nicotinamide improves kidney function in mice with acute kidney injury induced by ischemia/reperfusion or treatment with cisplatin [29,30]. In addition, nicotinamide prevents apoptosis in different cellular injury paradigms [31]. Consequently, one would expect that apoptotic treatments would increase nicotinamide consumption to protect HK-2 cells so that there should be a depletion of this vitamin in the culture medium, in which it is a normal component whose uptake by proximal tubular cells occurs through an unidentified transporter [32,33]. However, its levels were higher than in the medium from control cells (Appendix A); thus, the most likely explanation for this result is that the nicotinamide transporter is inhibited by cisplatin and, more strongly, by UV light exposure. This might facilitate their noxious effects on proximal tubular cells, which obviously would only be medically relevant in the case of cisplatin.

Creatine and spermine where two metabolites unequivocally identified whose content changed only between the ABs (Appendix A). Creatine and phosphocreatine provide an energy-buffering system that is essential for the maintenance of ATP supply in cells with high energy demands, such as proximal tubular cells (to note, just their Na^+^/K^+^-ATPase already consumes 80% of the O_2_ kidney supply in basal conditions) [34] Spermine is a polyamine and its enhanced catabolism, together with that of other polyamines, is critical to cisplatin-induced renal cell apoptosis in mice [35]. However, because there were no differences in the content of spermine and creatine between control HK-2 cells and the ABs, it is unlikely that they contribute to propagate apoptosis. In addition to spermine, another metabolite related with polyamines, 5’-methylthioadenosine, showed changes when compared to controls (Appendix A). 5’-methylthioadenosine is a sulfur-containing nucleoside present in all mammalian tissue and it is mainly produced from S-adenosylmethionine through the polyamine biosynthetic pathway, during the biosynthesis of polyamines [36]. 5’-methylthioadenosine has been shown to influence several critical responses of the cell, such as proliferation, differentiation, and regulation of gene expression, and to trigger apoptosis in cancer cells [37,38,39]. Therefore, 5’-methylthioadenosine might collaborate in the propagation of apoptosis to naïve HK-2 cells by ABs from cisplatin-treated cells (but not by ABs from UV light-treated cells because their content in 5’-methylthioadenosine was not statistically different from that found in control cells). Again, these results suggest that, depending on the ABs’ origin, there may be quantitative differences in their metabolite content, which may affect the degree to which ABs propagate apoptosis to naïve proximal tubular cells. It is even possible that the 5’-methylthioadenosine secreted by apoptotic cells to the culture medium may contribute to the propagation of apoptosis. If this was the case, because there are statistically significant differences between the levels of 5’-methylthioadenosine in culture medium from HK-2 cells treated with cisplatin and UV light, being higher in media from cisplatin-treated cell (Appendix A), there might be also quantitative differences in the degree in which the 5’-methylthioadenosine secreted by apoptotic cells contributes to the propagation of cell death to naïve HK-2 cells, depending on the nature of the apoptotic stimulus. Clearly, specific experiments are needed to assess the role of 5’-methylthioadenosine in the propagation of apoptosis.

Essential branched chain amino acids valine and leucine are present in the culture medium formulation from which they are taken up by proximal tubular cells by specific active transporters [40]. Treatment with cisplatin or exposure to UV light resulted in increased content in both amino acids in ABs (Appendix A) and culture medium (Appendix A), as compared to controls, particularly for the UV treated group. These results most likely imply that their uptake from medium and their use by HK-2 cells and/or by ABs are inhibited by the apoptotic stimuli. However, it is unlikely that the increase in the ABs content in valine and leucine may contribute to the propagation of apoptosis to naïve HK-2 cells because, to the best of our knowledge, there are no reports on an active role of these two amino acids in apoptosis. The same holds true for the N-(1-deoxy-1-fructosyl) derivatives from amino acids leucine, phenylalanine, and tyrosine (Appendix A), which are Amadori compounds. These compounds are N-substituted 1-amino-1-deoxyketoses formed during the initial stage of such reaction, Maillard reaction, in which amino acids readily react with reducing carbohydrates [41]. A screening of published articles on the matter does not provide any clear conclusion into the biological meaning of the derivatives of N-1-deoxy-1-fructosyl amino acid; therefore, their role in ABs-induced HK-2 cell apoptosis remains unknown. We can only affirm that these amino acids originate from the HK-2 cells because they are not present in the initial culture medium formulation.

We have previously found that ABs from HK-2 cells propagate apoptosis to naïve HK-2 cells independently of the nature of the initial apoptotic stimulus, i.e., treatment with cisplatin, exposure to UV light or nutrient deprivation, that generated the ABs [6]. Furthermore, prostaglandin E2 receptors and intracellular prostaglandin E2 were critical mediators for the cell death induced by the ABs generated by the different apoptotic stimuli [6]. Despite these similarities, the results presented in the current work indicate that there were stimulus-specific differences in several metabolites, either included into ABs or directly released to the extracellular medium by apoptotic cells. Accordingly, we hypothesize that these differences might correspondingly result in differences in the degree in which every one of these metabolites contribute to the total level of apoptosis induced by the different apoptotic stimuli. Unfortunately, there is not any available evidence in support of our view because of the absolute lack of studies on the apoptotic effects of the metabolites contained in ABs. Additional work is required to confirm that there are significant, stimulus-specific differences in metabolites abundancies that may propagate apoptosis to healthy proximal tubular cells because (i) it is not unlikely that the metabolite content in ABs may be affected by the isolation procedure (due to the lack of global consensus on standard protocols to extract and analyze EVs [42]), and (ii) further studies in cells other than HK-2 cells are needed to accurately verify our current data. The confirmation of our current results by this additional work has potential clinical applications: for instance, the analysis of the ABs metabolome in urine samples from cisplatin-treated patients might be a possible source for biomarker development in the context of early identification of patients at risk of cisplatin-induced acute kidney injury. 

## 3. Materials and Methods

### 3.1. Reagents and Solvents

All reagents and solvents that were used in this work were of analytical grade or higher. A Milli-Q System (Millipore, Bedford, MA, USA) was employed to obtain the water for the solutions. Acetonitrile, formic acid, and methanol were purchased from Thermo Fisher Scientific (Madrid, Spain).

To identify the molecular features selected, the standards: L-argininosuccinic acid lithium salt, creatine monohydrate, 5′-deoxy-5′-(methylthio)adenosine, L-homophenylalanine hydrochloride, 3-indoleacryltae, L-kynurenine, leucine, N-methyl-L-phenylalanine hydrochloride, nicotinamide, pyridoxine, spermine, and valine were acquired from Sigma Aldrich (Madrid, Spain).

### 3.2. Cell Culture

HK-2 cells were obtained in the American Type Culture Collection (ATCC) (Rockville, MD, USA) and were maintained in DMEM/F12 supplemented with 10% fetal bovine serum, 1% penicillin/streptomycin/amphoterycin B, 1% glutamine (Invitrogen, Carlsbad, CA, USA), and 1% insulin-transferrine-selenium (ThermoFisher, Grand Island, NY, USA). The culturing of these cells was conducted in a humidified 5% CO_2_ stove at 37 °C and were plated once cells reached 70–90% confluence.

### 3.3. Isolation of ABs from Apoptotic HK-2 Cells

Apoptosis of HK-2 cells was induced by incubation with 50 μM cisplatin (for 24 h) or by UV exposure (4000 µW/cm^2^ for 120 s). The medium of apoptotic HK-2 cells medium was collected without detaching the intact cells and was further separated from cell debris and dead cells by centrifugation (500× *g*, 5 min). This ABs-containing medium was centrifuged (5000× *g*, 10 min) once again to isolate the ABs pellet and the resulting medium was isolated and frozen at −70 °C until analyzed. After washing the ABs with PBS, the resulting PBS-free ABs pellets were frozen at −70 °C until further analyzed. As already demonstrated by García-Pastor et al. [6], the isolation process of ABs is highly robust. In that work, after the initial generation, the ABs were identified through several approaches: (a) specific staining with annexin V (Figure 1a from reference [6]), which demonstrated ABs origin from apoptotic cells; and (b) scanning electron microscopy, which showed that apoptotic cells and ABs display the characteristic morphology (Figure 1b from reference [6]). ABs size ranged between 0.57 μm and 2.27 μm (Figure 1a from reference [6] lower panel). Protein content was measured by the Pierce BCA-200 assay kit following indications of the manufacturer. The values for the protein content were 5.76, 1.22, and 1.04 mg/mL for control cells, ABs induced by cisplatin, and ABs induced by UV, respectively.

### 3.4. Optimized Sample Preparation Protocol

The procedure followed to prepare the samples analyzed was adapted from the one developed by our research group for the metabolomic profile analysis of HK-2 cells [10]. Briefly, 400 µL of 75% (v/v) MeOH in water was added to pelleted ABs and cells pellets while 300 µL of MeOH was added to 100 µL of extracellular fluid. The mixtures were vortexed for 30 s, kept in an ice-cold bath for 5 min, and centrifuged (14,000× *g*, 4 °C, 5 min). The resulting supernatant was transferred and evaporated until dryness for 3.5 h. Finally, the dried pellet was reconstituted in 100 µL of water, vortexed for 30 s, centrifugated at 14,000× *g* for 5 min, and placed into a glass insert for subsequent LC-MS analysis.

### 3.5. Liquid Chromatography-Mass spectrometry Metabolomics Analysis

To carry out the sample analysis, an 1100 series HPLC (Agilent Technologies, Waldbronn, Germany) coupled to a 6530 series quadrupole time-of-flight (QTOF) mass spectrometer (Agilent Technologies, Waldbronn, Germany) employing a Jet Stream orthogonal electrospray ionization (ESI) was employed. Agilent Mass Hunter Qualitative Analysis software (B.10.00) was used to control the LC-MS system and data acquisition. Analyses were performed in the positive ionization mode and a 25 mL Gastight 1000 Series Hamilton syringe (Hamilton Robotics, Bonaduz, Switzerland) on a NE-3000 pump (New Era Pump Systems Inc., Farmingdale, NY, USA) was employed to infuse continuously the solution with the reference ions (*m*/*z* 121.0509 (C_5_H_4_N_4_) and *m*/*z* 922.0098 (C_18_H_18_O_6_N_3_P_3_F_24_) at 15 µL/min in order to perform the mass correction.

The column selected for the analysis was a C18 Ascentis Express column (Sigma, St. Louis, MO, USA) of 100 × 2.1 mm i.d. dimensions (fused-core^®^ particles with 0.5 μm thick, porous shell, and an overall particle size of 2.7 μm). A 5 × 2.1 mm i.d. guard column of the same composition as the analytical column was employed. The column was placed in an oven and kept at 40 °C during the sample analysis. The injection volume of the samples was 10 μL and the mobile phase flow rate was 0.4 mL/min. Mobile phase A was water and mobile phase B was acetonitrile, both with 0.1% formic acid. The analyses were carried out using the following gradient: 5% B to 100% B in 30 min, 100% B for 5 min, returning to starting conditions (5% B) in 1 min, and keeping it for 15 min.

The QTOF parameters for the sample analysis were a capillary voltage of 3000 V with a nozzle voltage of 0 V, nebulizer pressure at 35 psig, sheath gas of jet stream of 6.5 L/min at 275 °C, and drying gas of 10 L/min at 275 °C. The cone voltage after the capillary (also known as fragmentator) was set at 100 V, while the skimmer and octapole voltages were at 750 V, respectively. MS analyses in the positive ESI mode were performed at a mass range set of *m*/*z* 70–1600 (extended dynamic range) in full scan resolution mode using a scan rate of 2 scans/s. MS/MS analyses were used to aid in the metabolite identification taking into account the [M+H]^+^ ions as precursor ions at the given retention time with a collision energy of 20 V. To ensure the repeatability in the system, several blanks and QC samples were analyzed at the beginning and at the end of the run and each in five samples.

### 3.6. Data Treatment and Analysis

Molecular Feature Extraction tool in Mass Hunter Qualitative Analysis (B.10.00 version) was used to create the molecular features, keeping the adducts in the positive ionization mode (H^+^, Na^+^, K^+^, and NH_4_^+^) and using a minimum of 12,000 counts (calculated as three times the signal-to-noise ratio). Agilent Mass Profiler Professional tool (B.15.1 version) was used to filter and align the data matrix. Retention time data were 0.1% with a window of 0.15 min. Mass tolerance was 20.0 ppm with a mass window of 2.0 mDa. SIMCA 14.0 (Umetrics, Umeå, Sweden) was employed to carry out the multivariate statistical analysis. To interpret the data correctly, log-transformation, Pareto scaling, and normalization against the protein content (see Section 3.3) were carried out previously to PCA and PLS-DA. Significant molecular features were chosen based on the so-called variable importance in the projection (VIP) values from the first component of the resulting PLS–DA models.

### 3.7. Identification of Metabolites

The identification step was performed taking into account the molecular features, which showed significant differences in the PLS-DA. In this line, the CEU Mass Mediator database from the Centre for Metabolomics and Bioanalysis (CEMBIO, Spain), which allows the search of metabolites in different databases, such as HMDB (http://www.hmdb.ca/ (accessed on 9 May 2022)), METLIN (https://metlin.scripps.edu (accessed on 16 May 2022)), LipidMaps (http://www.lipidmaps.org/ (accessed on 18 May 2022)), and KEGG (https://www.genome.jp/kegg/ (accessed on 23 May 2022)), simultaneously, was employed. The search was performed using the accurate mass values (with an error of 30 ppm). Only the metabolites, whose likelihood to be endogenous molecules in biological samples was high, were considered. Drugs or different exogenous compounds were discarded.

## 4. Conclusions

A RPLC-MS platform was employed to study the metabolic differences between ABs from HK-2 cells induced by cisplatin and UV light, where intracellular and extracellular fluids were analyzed in order to expand the metabolite coverage. PCA models showed good clustering between the experimental groups and PLS-DA models, and the resulting VIP values were used to dissect the differences between the experimental groups. Creatine and spermine (ABs fluid), nicotinamide (extracellular fluid) and valine, leucine, kynurenine, 5’-methylthioadenosine, and pyridoxine (both fluids), were identified unequivocally, while N-(1-Deoxy-1-fructosyl)-derivatives of (iso)leucine, phenylalanine and tyrosine were identified in a tentative manner. The results point towards different apoptotic mechanisms between the two different treatments. Among these changes we can highlight alterations in vitamin B6 and kynurenine pathways, and polyamine and branched chain amino acid related metabolism. Yet, future targeted studies should be conducted to confirm these results.

## Figures and Tables

**Figure 1 ijms-24-07237-f001:**
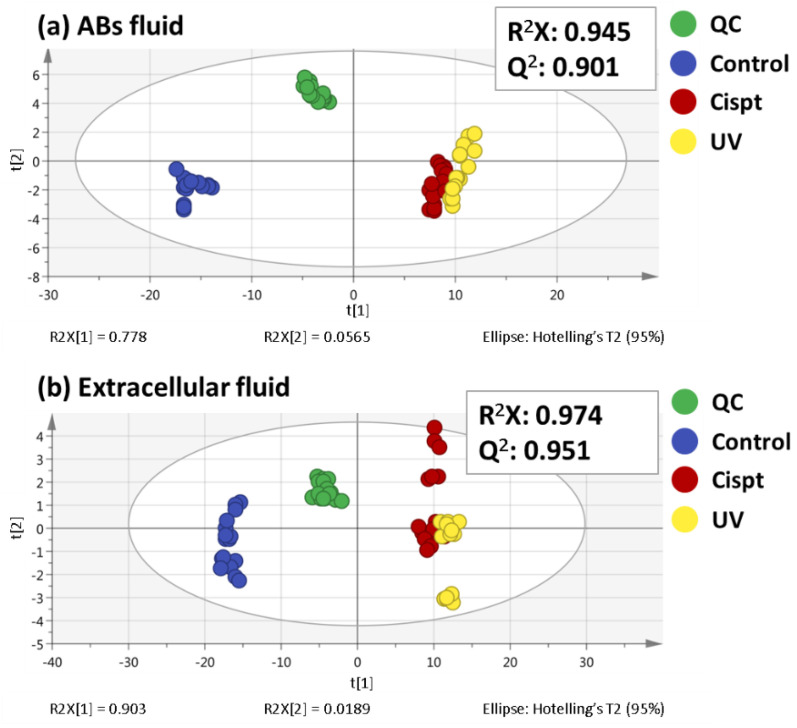
PCA score plots for both ABs fluid (**a**) and extracellular fluid (**b**) analyses.

**Figure 2 ijms-24-07237-f002:**
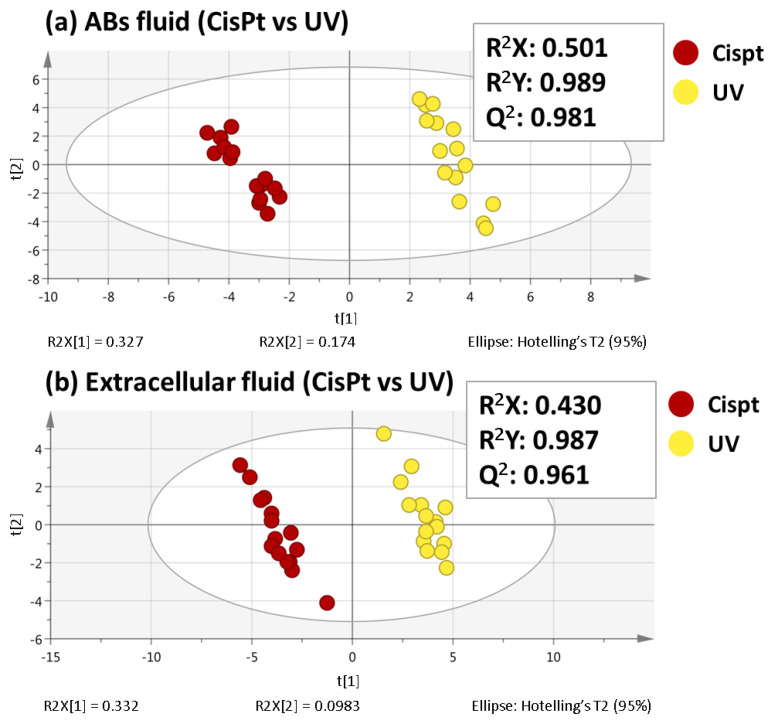
PLS-DA score plots (Cispt vs. UV) for both ABs fluid (**a**) and extracellular fluid (**b**) analyses.

**Figure 3 ijms-24-07237-f003:**
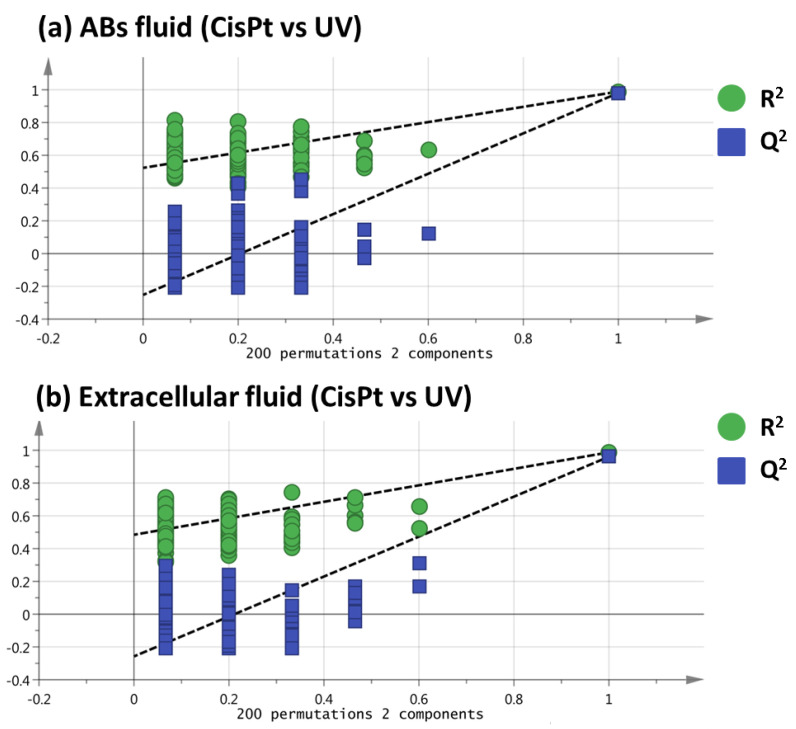
Permutation tests for the comparison of Cispt vs. UV of ABs fluid (**a**) and extracellular fluid (**b**) analyses.

**Table 1 ijms-24-07237-t001:** Metabolites identified in the Abs and extracellular fluid analyses by the RPLC-MS metabolomics platform.

#	RT (min)	Molecular Formula	Metabolite	Identification Level *	Monoisotopic Mass (Da)	Mass Error (ppm)	Main Fragments (MS/MS)	VIP (Trend CisPt vs. UV) **
**Intracellular fluid**								
1	0.5	C_10_H_26_N_4_	Spermine	1	202.2156	1	112.1105, 84.0804, 129.1387	1.3 (↑)
2	0.7	C_4_H_9_N_3_O_2_	Creatine	1	131.0663	24	90.0546, 87.0584	1.2 (↓)
3	0.8	C_5_H_11_NO_2_	Valine	1	117.0767	20	72.0794, 55.0533	1.2 (↓)
4	0.8	C_8_H_11_NO_3_	Pyridoxine	1	169.0727	7	134.0592, 152.0687	2.2 (↑)
5	0.8	C_15_H_21_NO_8_	N-(1-deoxy-1-fructosyl)tyrosine	2	343.1250	5	308.1066, 280.1201, 326.1201	3.5 (↓)
6	0.9	C_12_H_23_NO_7_	N-(1-deoxy-1-fructosyl)-(iso)leucine	2	293.1457	6	230.1371, 258.1314, 132.0993	3.7 (↓)
7	0.9	C_6_H_13_NO_2_	Leucine	1	131.0940	5	86.0948	2.0 (↓)
8	1.2	C_10_H_12_N_2_O_3_	Kynurenine	1	208.0843	2	146.0589, 94.0619, 118.0618, 74.0236, 174.0538, 120.0408	1.3 (↓)
9	1.3	C_15_H_21_NO_7_	N-(1-deoxy-1-fructosyl)phenylalanine	2	327.1310	2	292.1183, 310.1264, 166.0868, 178.0849	3.4 (↓)
10	2.5	C_11_H_15_N_5_O_3_S	5’-Methylthioadenosine	1	297.0889	2	136.0613	3.0 (↑)
**Extracellular fluid**								
11	0.8	C_6_H_6_N_2_O	Nicotinamide	1	122.0463	14	80.0486, 78.0337	1.5 (↓)
12	0.8	C_5_H_11_NO_2_	Valine	1	117.0764	22	72.0808, 58.0648, 59.0730	1.2 (↓)
13	0.8	C_8_H_11_NO_3_	Pyridoxine	1	169.0727	7	134.0587, 152.0695	2.0 (↓)
14	0.8	C_15_H_21_NO_8_	N-(1-deoxy-1-fructosyl)tyrosine	2	343.1225	12	308.1172, 280.0972, 326.1222	2.2 (↓)
15	0.9	C_12_H_23_NO_7_	N-(1-deoxy-1-fructosyl)-(iso)leucine	2	293.1463	4	230.1353, 258.1304, 276.1415	2.4 (↓)
16	0.9	C_6_H_13_NO_2_	Leucine	1	131.0937	7	86.0950	1.7 (↓)
17	1.2	C_10_H_12_N_2_O_3_	Kynurenine	1	208.0849	1	146.0600, 94.0651, 118.0642, 74.0233, 174.0503, 120.0429	1.3 (↓)
18	1.3	C_15_H_21_NO_7_	N-(1-deoxy-1-fructosyl)phenylalanine	2	327.1312	2	292.1177, 310.1231, 166.0857, 178.0860	1.8 (↓)
19	2.5	C_11_H_15_N_5_O_3_S	5’-Methylthioadenosine	1	297.0891	2	136.0620	2.3 (↑)

* Identification according to the Metabolomics Standard Initiative (MSI) guidelines [16]: level 1 (unequivocal identification matching retention time, accurate mass, and MS/MS of that of the standard solution); level 2 (identification based on accurate mass and fragmentation, compared to available reference database). ** ↑: The metabolite (on average) is more abundant in ABs induced by cisplatin; ↓: The metabolite (on average) is less abundant in ABs induced by cisplatin.

## Data Availability

Datasets of the metabolomic sequences are available with the manuscript.

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
