# Peer review of "Exploring the Metabolic Differences between Cisplatin- and UV Light-Induced Apoptotic Bodies in HK-2 Cells by an Untargeted Metabolomics Approach"

_ijms, 2023, doi:10.3390/ijms24087237_

Round 1

Reviewer 1 Report

This study investigated non-targeted metabolomic analysis on ABs and medium from apoptotic HK-2 cells induced with cisplatin or exposure to UV light.

These results present scientific evidence that there was an increase in several metabolites in Abs and the extracellular medium of apoptotic cells, which are potentially relevant for the propagation of apoptosis to naïve proximal tubular cells.

In addition, it is necessary to present research results or references in cells other than HK-2 cells to accurately verify the research results.

Depending on the degree of apoptosis, it is necessary to confirm changes in metabolites such as pyridoxine, kynurenine, and nicotinamide, and it is required to add relevant information to the ‘dicussion’.

Author Response

This study investigated non-targeted metabolomic analysis on ABs and medium from apoptotic HK-2 cells induced with cisplatin or exposure to UV light.

These results present scientific evidence that there was an increase in several metabolites in Abs and the extracellular medium of apoptotic cells, which are potentially relevant for the propagation of apoptosis to naïve proximal tubular cells.

We would like to thank this Reviewer for her/his suggestions that have helped us to improve the quality of this research work.

Point 1: In addition, it is necessary to present research results or references in cells other than HK-2 cells to accurately verify the research results.

Response 1: There are no references on the apoptotic effects of metabolites contained in ABs because the field of the role of ABs in pathology is in its infancy. To the best of our knowledge, there are only a few references regarding their implication in tumorigenesis, autoimmunity, inflammation and viral infection. These references have been now included in the reviewed version of the manuscript (please see Discussion). We understand the reviewer’s point of view that our current results would be reinforced through presenting additional results in cells other than HK-2 cells and, accordingly, this has now been considered in the Discussion. However, our study was only conceived as a proof-of-concept to show that there are stimulus-specific differences in several metabolites abundancies - either included into ABs or directly released to the extracellular medium by apoptotic cells- that might correspondingly result in differences in the degree in which each of these metabolites contribute to the total level of apoptosis induced by the different apoptotic stimuli.

Point 2: Depending on the degree of apoptosis, it is necessary to confirm changes in metabolites such as pyridoxine, kynurenine, and nicotinamide, and it is required to add relevant information to the ‘dicussion’.

Response 2: Our hypothesis was that, although different stimuli induce HK-2 cell apoptosis to a similar extent (as previously published in Biochimica et Biophysica Acta (BBA) - Molecular Basis of Disease 2019, 1865, 2504–2515 (ref.6 in the manuscript)), there are qualitative and/or quantitative stimulus-specific differences in metabolites that might influence the degree in which they contribute to apoptosis. Therefore, what we meant with “…whether the type of apoptotic stimulus gives rise to significant differences in these metabolites that might influence the propagation of apoptosis” was that we wanted to identify differences in metabolites involved in the propagation of apoptosis, as already indicated in the abstract. We did not mean that these differences will result in differences in the extension of apoptosis induced by ABs because we knew from our previous work (ref. 6) that the level of apoptosis was the same regardless the nature of the apoptotic stimulus. We rather meant that, as previously mentioned, there are stimulus-specific differences in several metabolites abundancies that might correspondingly result in differences in the degree in which every one of these metabolites contribute to the total level of apoptosis induced by the different apoptotic stimuli. The Discussion has been modified to help to understand these concepts (Lines 179-188 and 298-309).

Reviewer 2 Report

This is a nice paper. However, I have some comments. The findings from this paper are excellent and worthy to review. This manuscript contained some questions described below. I think this paper is interesting, this review contributes to future's clinical medicine largely. I have some questions from a point of view of clinical medicine. This paper is a basic study and contains very meaningful content, but how can the authors make use of the results obtained in this study from a clinical perspective? Please tell us. The reviewer wonders if acute tubular necrosis in an ischemia-reperfusion model could be an event similar to this study clinically, but clinically we often experience tubular cell regeneration afterwards. However, clinically, we often experience tubular cell regeneration afterwards. The present results showing apoptosis in tubular cells should be understood as a reversible event. Is this correct? Please tell me.

Author Response

This is a nice paper. However, I have some comments. The findings from this paper are excellent and worthy to review. This manuscript contained some questions described below. I think this paper is interesting, this review contributes to future's clinical medicine largely. I have some questions from a point of view of clinical medicine.

We would like to thank this Reviewer for considering that our “findings are excellent” and that “this manuscript contributes to future´s clinical medicine largely”.

Point 1: This paper is a basic study and contains very meaningful content, but how can the authors make use of the results obtained in this study from a clinical perspective? Please tell us.

Response 1: We have reviewed ("apoptotic bodies"[Title/abstract]) AND (metabolomics[Title/abstract]) and found no results. This illustrates that our understanding of the comprehensive effects of metabolites in these nanoparticles is currently limited and it is in its infancy. Furthermore, to the best of our knowledge, even if we focus only on the role of ABs in pathology, there are only a few refererences regarding the implication of ABs in tumorigenesis, autoimmunity, inflammation and viral infection. These few references have been now included in the reviewed version of the manuscript (please see Discussion). Regarding the clinical perspective of our results, we might hypothesize that the analysis of the ABs metabolome in urine samples from cisplatin-treated patients may be interesting as a possible source for biomarker development in the context of early identification of patients at risk of cisplatin-induced acute kidney injury. However, our current results are too preliminary to support the application of ABs metabolome to this aim. In any case, we have included the concepts on the clinical perspective of our results in the Discussion (Lines 310-318).

Point 2: The reviewer wonders if acute tubular necrosis in an ischemia-reperfusion model could be an event similar to this study clinically, but clinically we often experience tubular cell regeneration afterwards. However, clinically, we often experience tubular cell regeneration afterwards. The present results showing apoptosis in tubular cells should be understood as a reversible event. Is this correct? Please tell me.

Response 2: We have shown in Biochimica et Biophysica Acta (BBA) - Molecular Basis of Disease 2019, 1865, 2504–2515 (please see reference 6 in the manuscript) that ABs from cisplatin-treated cells are capable of reproducing two early noxious effects of the drug on HK-2 cells, namely apoptosis and inhibition of cell proliferation. However, in a later stage, dying cells promote wound repair through tubular cell proliferation (ref. 6) as previously observed in rats with cisplatin-induced acute kidney injury (S. Nakagawa et al T. Extracellular nucleotides from dying cells act as molecular signals to promote wound repair in renal tubular injury, AJP Ren. Physiol. 307 (2014) F1404–F1411). This is similar to the findings in the ischemia-reperfusion model. In fact, in our previous work (ref 6), we hypothesized that the ABs responsible for the propagation of cell proliferation should be generated in a second stage by ABs-exposed apoptotic cells. Accordingly, ABs generated by cisplatin-treated HK-2 cells induced the release of 2nd generation ABs that stimulated HK-2 cell proliferation but not apoptosis. These results indicated that ABs do not have always the same effects on recipient cells. This suggests that the content of ABs in molecules that mediate intercellular communication is dependent on the stimulus that triggers apoptosis. Clearly, understanding the different mechanisms through which ABs induce apoptosis or cell proliferation is a very relevant issue and a great challenge because ABs transport many molecules (such are cytokines, bioactive lipids, metabolites, proteins, mRNA and miRNAs) that play a critical role in intercellular communication.

We cannot include these concepts in the reviewed version of the manuscript because we have not analysed 2nd generation ABs.

Reviewer 3 Report

This manuscript is very interesting and well written. To date, there is not much information available about the impact of apoptotic bodies (ABs) in propagating cell death so, understanding of the impact of different apoptosis stimulus on the apoptotic bodies content and their way of action in HK-2 cells is very important in the context of kidney injuries and other diseases. 

minor revisions:

- the authors must increase the figures quality and size. They should also adjust the figures size in the manuscript, for example figures  2 A) and B) aren't the same size.

Author Response

This manuscript is very interesting and well written. To date, there is not much information available about the impact of apoptotic bodies (ABs) in propagating cell death so, understanding of the impact of different apoptosis stimulus on the apoptotic bodies content and their way of action in HK-2 cells is very important in the context of kidney injuries and other diseases.

We thank this Reviewer for considering that the manuscript is “well written” and for highlighting the relevance of our work.

Point 1: The authors must increase the figures quality and size. They should also adjust the figures size in the manuscript, for example figures  2 A) and B) aren't the same size.

Response 1: According to the reviewer’s comment, the quality and size of the figures have been increased and Figures 2 a) and b) have the same size now.

Reviewer 4 Report

The manuscript by Bernardo-Bermejo analyses the metabolic differences between apoptotic bodies released by HK-2 cells, upon induction to apoptosis either by cisplatin or UV light. Although the topic may be potentially of interest, the manuscript is unclear, as well as the main findings. In addition, some methodological issues need to be addressed.

Major points

The conclusion of the abstract is weak. What do authors mean for “… different apoptotic mechanisms between the two different treatments (cisplatin vs UV light exposed ABs).” Authors should be more precise and explicit what are the main differences, also in consideration that the apoptotic process has been widely characterized.

The rationale of the study is confused: authors state that they “…previously found that ABs from human renal proximal tubular epithelial HK-2 cells propagate apoptosis to naïve HK-2 cells independently of the type of the initial apoptotic stimulus”, but then they decide to investigate “…whether the type of apoptotic stimulus gives rise to significant differences in these metabolites that might influence the propagation of apoptosis”. If the output was similar, i.e. apparently the apoptotic process induced by different stimuli was similar, why did authors decided to investigate metabolic differences?

The manuscript is written in an unclear and difficult to catch manner: the statistical analysis section precedes the section dedicated to the explanation of metabolic species identification, but the statistical analysis is actually performed on the identified species, so the order should be the opposite, or at least some additional explanation should be given.

Authors did not provide a section aimed at the characterization of apoptotic bodies from different stimuli (yield, morphology, size distribution, presence, or absence of markers). A specific section should be provided, it is not possible just to refer to a previous study).

In the 2.4 section, authors claim that “…Among the changes potentially relevant are the differences in the content of pyridoxine and kynurenine between the two types of ABs, i.e. cisplatin-treated vs UV light exposed-cells, both in ABs content (Figures S1) and in the extracellular 184 fluid (Figure S2).”. However, it is inconsistent to begin a section discussing data exclusively reported in the supplementary material.

The statistical analysis (PCA, PLS-DA) was carried out using 15 samples. However, authors stated that “Each of these two metabolomics sequences consisted of five biological replicates per group and each being analysed three times (a total of 45 sample injections per run)”. However, it is not correct to report the analysis in triplicate as 3 different samples were analysed: the correct number of replicates is 5 and not 15.

Minor points

Line 43 Please do not begin a sentence with “that”, such as “That is why the study of 43 EVs has become a trending topic for biomarker searching [2].”

Line 44 Authors mentioned the recently identified exomeres , exosomes and microvesislces (“EVs can be classified as exomeres, exosomes, microvesicles, and apoptotic bodies (ABs),…”), but then they do not provide at least a definition for exomeres, exosomes and microvesicles

Author Response

The manuscript by Bernardo-Bermejo analyses the metabolic differences between apoptotic bodies released by HK-2 cells, upon induction to apoptosis either by cisplatin or UV light. Although the topic may be potentially of interest, the manuscript is unclear, as well as the main findings. In addition, some methodological issues need to be addressed.

We would like to thank this Reviewer for her/his suggestions that have helped us to improve the quality of this research work. Please, find below a detailed response to each comment.

 Point 1: The conclusion of the abstract is weak. What do authors mean for “… different apoptotic mechanisms between the two different treatments (cisplatin vs UV light exposed ABs).” Authors should be more precise and explicit what are the main differences, also in consideration that the apoptotic process has been widely characterized.

Response 1: The conclusion of the abstract has been modified (Lines 30-32).

Point 2: The rationale of the study is confused: authors state that they “…previously found that ABs from human renal proximal tubular epithelial HK-2 cells propagate apoptosis to naïve HK-2 cells independently of the type of the initial apoptotic stimulus”, but then they decide to investigate “…whether the type of apoptotic stimulus gives rise to significant differences in these metabolites that might influence the propagation of apoptosis”. If the output was similar, i.e. apparently the apoptotic process induced by different stimuli was similar, why did authors decided to investigate metabolic differences?

Response 2: As mentioned in our answer to Reviewer 2 comments, our hypothesis was that, although different stimuli induce HK-2 cell apoptosis to a similar extent, there are qualitative and/or quantitative stimulus-specific differences in metabolites that might influence the degree in which they contribute to apoptosis. Therefore, what we meant with “…whether the type of apoptotic stimulus gives rise to significant differences in these metabolites that might influence the propagation of apoptosis” was that we wanted to identify differences in metabolites involved in the propagation of apoptosis, as already indicated in the abstract. We did not mean that these differences will result in differences in the extension of apoptosis induced by ABs because we knew from our previous work published in Biochimica et Biophysica Acta (BBA) - Molecular Basis of Disease 2019, 1865, 2504–2515 that the level of apoptosis was the same independently of the nature of the apoptotic stimulus. We rather meant that there are quantitative and/or qualitative stimulus-specific differences in several metabolites that might correspondingly result in differences in the degree in which every one of these metabolites contribute to the total level of apoptosis induced by the different apoptotic stimuli.

We understand from the reviewer’s comment that our sentence leads to missunderstand our hypothesis and we have corrected it in the reviewed version of the manuscript (Lines 56-60).

Point 3: The manuscript is written in an unclear and difficult to catch manner: the statistical analysis section precedes the section dedicated to the explanation of metabolic species identification, but the statistical analysis is actually performed on the identified species, so the order should be the opposite, or at least some additional explanation should be given.

Response 3: According to the untargeted metabolomics workflow the statistical analysis step precedes the identification of the features. When the statistical analysis is carried out, all metabolic features are unknown, we only know the exact mass and the retention time of each one. Based on the statistical analysis and, on the variable important in the projection (VIP) values obtained from PLS-DA models, we selected those statistically significant metabolites and then the identification step took place. For this reason, we consider that the order proposed in the manuscript is correct. Anyway, to clarify this procedure, a better explanation has been included in lines 152, 153.

Point 4: Authors did not provide a section aimed at the characterization of apoptotic bodies from different stimuli (yield, morphology, size distribution, presence, or absence of markers). A specific section should be provided, it is not possible just to refer to a previous study).

Response 4: The authors respectfully disagree with the reviewer’s point of view: the aim of the study is not an extracellular vesicle-focused characterization of ABs (there are plenty of studies on the field) but rather a metabolomic-focused characterization of ABs in relation to the apoptotic ability of their metabolites because, to the best of our knowledge, this is the first time that metabolomics is applied to fulfill this aim. In this context, the authors think that the extracellular vesicle-focused characterization of ABs proposed by the reviewer, though adding interesting supplementary information, would not improve the conclusions of our study.

Point 5: In the 2.4 section, authors claim that “…Among the changes potentially relevant are the differences in the content of pyridoxine and kynurenine between the two types of ABs, i.e. cisplatin-treated vs UV light exposed-cells, both in ABs content (Figures S1) and in the extracellular 184 fluid (Figure S2).”. However, it is inconsistent to begin a section discussing data exclusively reported in the supplementary material.

Response 5: It has been modified in the corrected version of the manuscript (Lines 179-188).

Point 6: The statistical analysis (PCA, PLS-DA) was carried out using 15 samples. However, authors stated that “Each of these two metabolomics sequences consisted of five biological replicates per group and each being analysed three times (a total of 45 sample injections per run)”. However, it is not correct to report the analysis in triplicate as 3 different samples were analysed: the correct number of replicates is 5 and not 15.

Response 6: The sentence “a total of 45 sample injections per run” has been removed in line 95 and we haved stressed that each sample was analyzed in triplicate as analytical replicates.

Point 7: Line 43 Please do not begin a sentence with “that”, such as “That is why the study of 43 EVs has become a trending topic for biomarker searching [2].”

Response 7: The sentence in line 42 has been modified according to the reviewer’s suggestion.

Point 8: Line 44 Authors mentioned the recently identified exomeres, exosomes and microvesislces (“EVs can be classified as exomeres, exosomes, microvesicles, and apoptotic bodies (ABs),…”), but then they do not provide at least a definition for exomeres, exosomes and microvesicles

Response 8: Following the reviewer´s comment, additional information about the classification of extracellular vesicles has been included in lines 43-46.

Round 2

Reviewer 4 Report

The manuscript by Bernardo-Bermejo has improved most weaknesses of the previous version. However, about Point 2, it is still unclear on what basis authors hypothesized that there might be there qualitative and/or quantitative stimulus-specific differences in metabolites that might influence the degree in which they contribute to apoptosis. Are there any previous studies suggesting this evidence? If there are any, suitable references should be added. About point 4, the isolation of EVs is still a complex problem and their content, including metabolites, is strongly affected by the isolation procedure. When authors specified that metabolites may be either included into ABs or directly released to the extracellular medium by apoptotic cells (line 299), they should also explicit this issue.

Author Response

The manuscript by Bernardo-Bermejo has improved most weaknesses of the previous version.

Point 1: However, about Point 2, it is still unclear on what basis authors hypothesized that there might be there qualitative and/or quantitative stimulus-specific differences in metabolites that might influence the degree in which they contribute to apoptosis. Are there any previous studies suggesting this evidence? If there are any, suitable references should be added.

Response 1: We acknowledge the reviewer for his/her comment because we share his/her concerns. As stated in the first sentence of the Biological Interpretation section, “There are no references on the apoptotic effects of metabolites contained in ABs because the field of the role of ABs in pathology is in its infancy”. Therefore, our view proposing that “there might be there qualitative and/or quantitative stimulus-specific differences in metabolites that might influence the degree in which they contribute to apoptosis”remains hypothetical and we state so in the second revision of the manuscript (please see blue text in the ABSTRACT (lines 31, 32) and Biological interpretation (lines 302, 303, 305-312, 313,314).

Point 2: About point 4, the isolation of EVs is still a complex problem and their content, including metabolites, is strongly affected by the isolation procedure. When authors specified that metabolites may be either included into ABs or directly released to the extracellular medium by apoptotic cells (line 299), they should also explicit this issue.

Response 2: We agree with the reviewer’s view. We address the issue of the dependency of the metabolite content in ABs on the isolation procedure in the Biological interpretation (please see blue text (lines 302, 303, 305-312, 313,314).    
